# TerC proteins function during protein secretion to metalate exoenzymes

Bixi He [1], Ankita J. Sachla [1] & John D. Helmann [1] ✉

Cytosolic metalloenzymes acquire metals from buffered intracellular pools. How exported metalloenzymes are appropriately metalated is less clear. We provide evidence that TerC family proteins function in metalation of enzymes during export through the general secretion (Sec-dependent) pathway. *Bacillus subtilis* strains lacking MeeF(YceF) and MeeY(YkoY) have a reduced capacity for protein export and a greatly reduced level of manganese (Mn) in the secreted proteome. MeeF and MeeY copurify with proteins of the general secretory pathway, and in their absence the FtsH membrane protease is essential for viability. MeeF and MeeY are also required for efficient function of the $Mn^{2+}$-dependent lipoteichoic acid synthase (LtaS), a membrane-localized enzyme with an extracytoplasmic active site. Thus, MeeF and MeeY, representative of the widely conserved TerC family of membrane transporters, function in the co-translocational metalation of $Mn^{2+}$-dependent membrane and extracellular enzymes.

Metal ions are essential for life, in large part due to their roles as cofactors for enzymes where they can serve as an electrophilic center or redox catalyst[1,2]. Metalloenzymes most often function with a specific metal that is acquired during protein folding or by binding of metal to an already folded apo-protein. Some metals bind proteins with high affinity and may therefore exchange slowly if at all. Others bind to metals more loosely and exchange frequently. Typical metal affinities are summarized in the Irving-Williams series: Mn(II)<Fe(II)<Co(II)<Ni(II)<Cu(II)>Zn(II)[3]. The most abundant metals in the cytosol (Mn, Fe, and Zn) are generally present as divalent ions and are referred to here without reference to ionic state. Cytosolic enzymes acquire metals from a buffered pool[4,5], with Mn and Fe at low micromolar levels[6,7]. High affinity metals such as Cu and Zn are buffered to subnanomolar levels of free ions. This implies that enzymes acquire metals by exchange reactions with bound metal-ligand complexes, and these may include protein metallochaperones that help ensure insertion of the correct metal[8,9].

Metalation of enzymes with active sites external to the cell membrane is not as well understood. Zn-requiring enzymes may acquire this ion from the environment[10,11]. However, metalation of enzymes that require lower affinity metals is more problematic. For example, if a Mn-requiring enzyme is secreted from the cell without an associated metal ion it can easily be mismetallated by Cu or Zn[10]. One solution is to metalate the protein inside the cell and secrete the folded metalloprotein through the TAT-dependent secretion system[10]. However, the TAT system cannot be used for integral membrane proteins that require metal cofactors. In addition, many metalloproteins are exported in an unfolded state lacking bound metals through the SecYEG-dependent general secretion pathway. How these exported proteins are properly metalated is not always clear.

TerC proteins (Pfam03741) are poorly understood membrane proteins originally described in bacteria and implicated in resistance to the toxic anion tellurite[12], although the mechanism of resistance is still elusive[13]. More recently, TerC proteins have been implicated in Mn export[14,15]. In *Bacillus subtilis*, a mutant strain missing the MneP and MneS cation diffusion facilitator proteins is highly sensitive to Mn intoxication[16]. This defect can be suppressed by increased expression of a TerC homolog, YceF[15]. Conversely, an *mneP mneS yceF* triple mutant has enhanced intracellular accumulation of Mn in cells after Mn shock[15]. These and other results suggest that YceF and its paralog YkoY have Mn efflux activity[15]. However, this activity is minor when compared to that of MneP and MneS. Similarly, an *Escherichia coli* TerC homolog, Alx, has recently been implicated in Mn efflux[17] and TerC family proteins are linked to metal homeostasis in several systems[14]. However, the precise role of TerC proteins in Mn homeostasis is unclear.

[1]Department of Microbiology, Cornell University, 370 Wing Hall, 123 Wing Drive, Ithaca, NY 14853-8101, USA. ✉e-mail: jdh9@cornell.edu

Here, we report that the *B. subtilis* TerC proteins MeeF(YceF) and MeeY(YkoY) are involved in metalation of exoenzymes with Mn, an important cofactor for diverse enzymes[18,19]. Mutants lacking these two TerC proteins were defective in protein secretion and in synthesis of membrane-associated lipoteichoic acids (LTA), which depends on the Mn-dependent LtaS enzyme. Proteomic and genetic studies indicated that TerC proteins interact with the secretosome, suggestive of a role in co-translocational protein metalation. Consistently, the FtsH protease, critical for clearing jammed translocons in the membrane[20], was essential in a *meeF meeY* double mutant, and overexpression of FtsH improved fitness of the *meeF meeY* mutant. Our results implicate TerC proteins as accessory subunits of the secretosome that mediate the metalation of Mn-dependent exoenzymes. A similar biochemical role may explain phenotypes resulting from mutations in the related plant[21], yeast[22], and human[23] proteins.

## Results

### Cells lacking MeeF and MeeY are defective in production of extracellular proteases

Evidence that the TerC proteins have functions distinct from protection against Mn intoxication first emerged from a simple observation. We noted that a double mutant (Δ*meeF* Δ*meeY;* designated FY) lacking both MeeF (formerly YceF) and MeeY (formerly YkoY) displayed a large (70%) decrease in colony size on LB agar (Fig. 1a, Supplementary Fig. 1a). The reduction in colony size for the *meeF* and *meeY* single mutants was much less dramatic. This suggests that MeeF and MeeY (~40% aa identity) have overlapping functions important for fitness during growth on solid medium. Loss of a third TerC homolog, YjbE, did not affect fitness under these growth conditions (Fig. 1a), consistent with prior studies demonstrating that it is expressed primarily during sporulation[24]. In contrast with their growth defect on solid LB

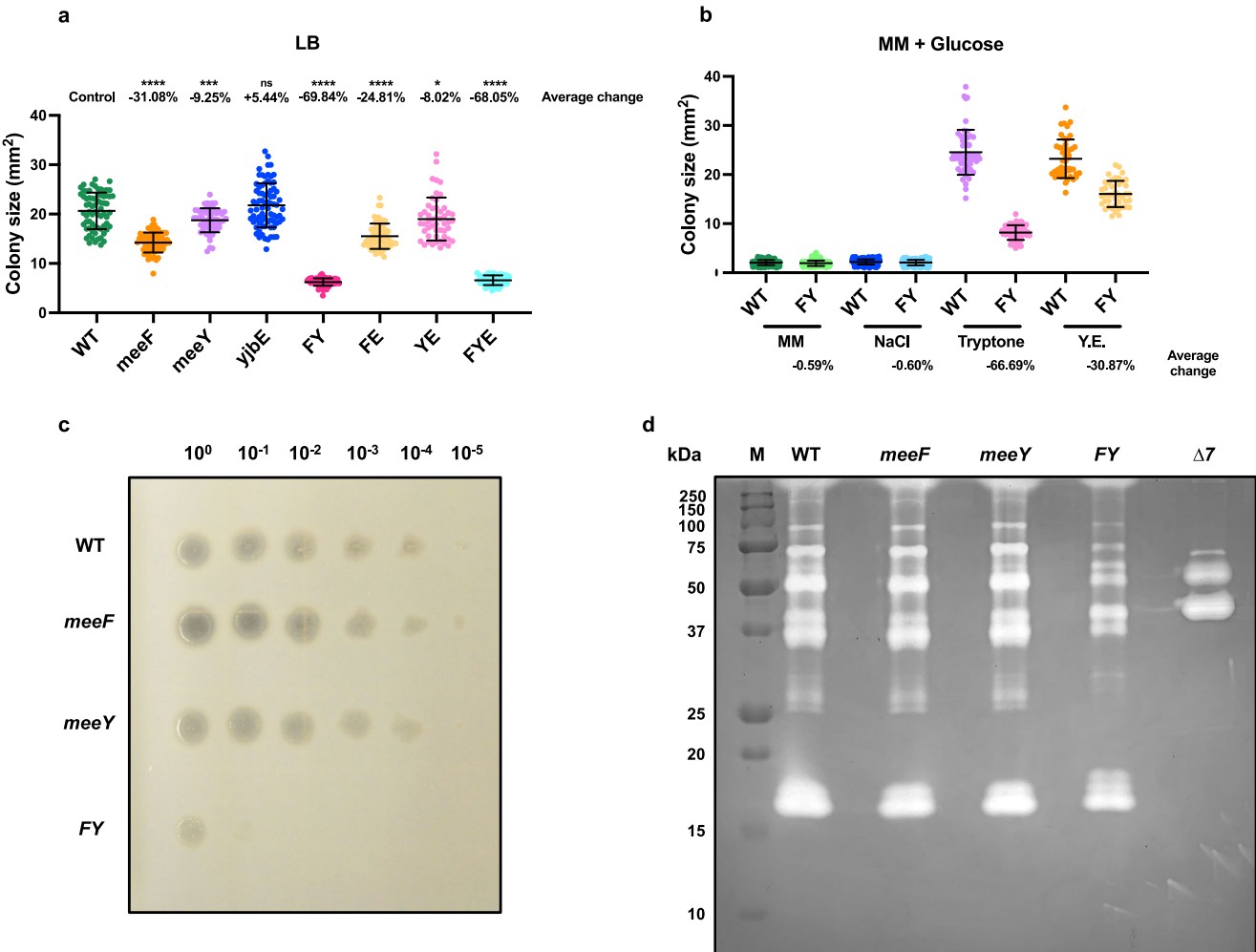

**Fig. 1 | MeeF and MeeY are required for efficient secretion of feeding proteases to access nutrients in tryptone. a** Colony sizes of WT, *meeF, meeY, yjbE*, FY, FE *(meeF yjbE)*, YE *(meeY yjbE)* and FYE *(meeF meeY yjbE)* strains on LB agar. **b** Colony sizes of WT and FY mutant on a defined glucose-minimal media (MM). MM agar plates were made with and without NaCl, tryptone, or yeast extract (Y.E.). In (**a**) and (**b**), agar plates with well isolated colonies were imaged after 24 h at 37 ˚C, and sizes measured using ImageJ. Isolated colonies from three independent cultures were included in the measurements for each strain and data are presented as mean ± standard deviation. In (**a**), average changes were calculated as "change = (sample - control) / control * 100%". Sample sizes were: WT, $n = 70$; *meeF*, $n = 68$; *meeY*, $n = 57$; *yjbE*, $n = 77$; FY, $n = 78$; FE, $n = 63$; YE, $n = 51$; FYE, $n = 48$. *P* value was calculated using Welch's *t* test, two-tailed, and compared to WT *meeF*, ****$p < 0.0001$; *meeY*,

****$p = 0.0007$; *yjbE*, ns$p = 0.0986$; FY, ****$p < 0.0001$; FE, ****$p < 0.0001$; YE, *$p = 0.0303$; FYE, ****$p < 0.0001$. In (**b**), average changes were calculated as in (**a**) and WT is the control. Sample size for MM plates, WT, $n = 317$; FY, $n = 260$; for NaCl, WT, $n = 200$; FY, $n = 186$; for Tryptone, WT, $n = 54$; FY, $n = 68$; for Y.E., WT, $n = 47$; FY, $n = 43$. **c** Protease activity on 5% milk agar plates. Serial dilutions ($10^0$–$10^{-5}$) of cells were inoculated on plates. The image is representative of three independent experiments. **d** Extracellular protease activities in the supernatants were detected by gelatin zymography. Supernatants were collected from overnight cultures with the same cell number. M, protein molecular weight marker; Δ7, mutant lacking seven extracellular proteases (Δ*aprE*, Δ*nprE*, Δ*nprB*, Δ*bpr*, Δ*epr*, Δ*mpr*, Δ*vpr*). The image is representative of three independent experiments. Source data are provided as a Source Data file.

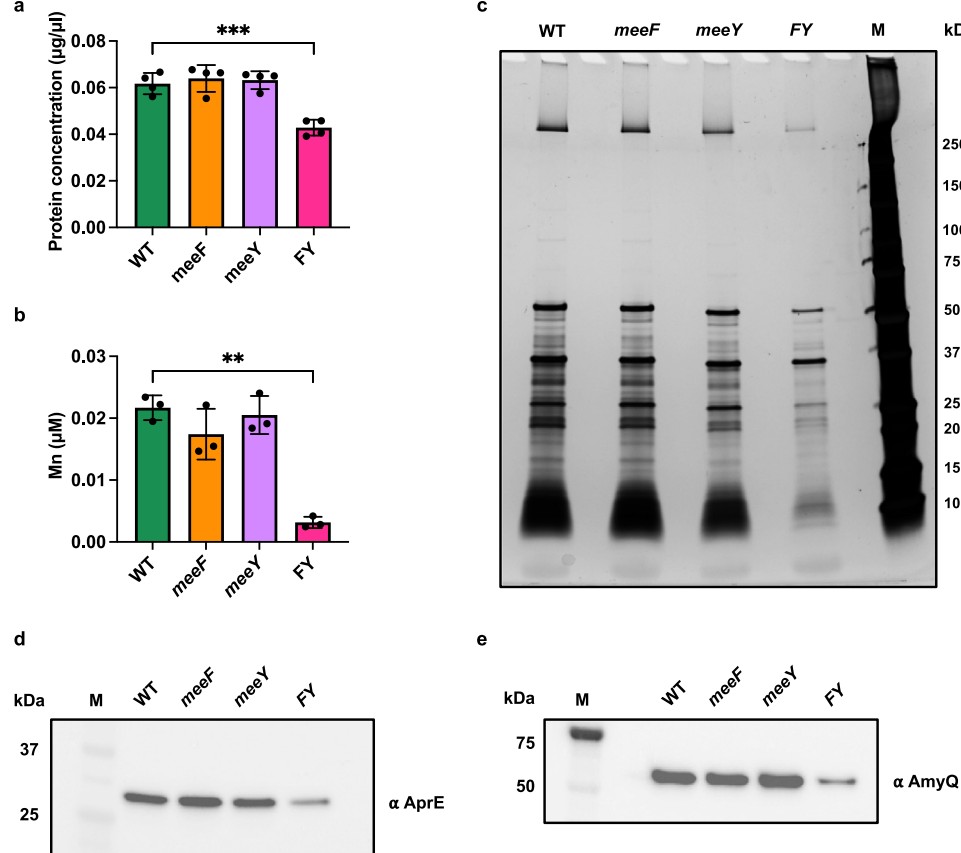

**Fig. 2 | FY mutants have a generalized secretion defect. a** FY mutants (but not the *single meeF and meeY* mutants) have reduced levels of secreted proteins in the spent medium (supernatant fraction) after overnight culture. Data are presented as mean ± standard deviation. ***$p$ = 0.0008, $P$ values were calculated using Welch's $t$ test, two-tailed. $n$ = 4 independent experiments. **b** FY mutants have dramatically reduced levels of Mn in the spent medium after overnight growth as monitored by ICP-MS analysis. Data are presented as mean ± standard deviation. **$p$ = 0.001. $P$ values were calculated using Welch's $t$ test, two-tailed. $n$ = 3 independent experiments. **c** Silver-stained SDS-PAGE analysis showing reduced levels of extracellular proteins in the supernatant from the FY strain compared to WT and the single

mutant strains after overnight growth (representing the same final culture density; Supplementary Fig. 1b). M, protein molecular weight marker. The image is representative of three experiments. **d** Western blot analysis of AprE-FLAG in supernatants. The image is representative of three independent experiments. Pellet samples and Ponceau-stained images (to demonstrate equal loading) are shown in Supplementary Fig. 4a. **e** Western blot analysis of heterologous AmyQ-His secretion. The image is representative of three independent experiments. Pellet samples and Ponceau-stained images are shown in Supplementary Fig. 4b. Source data are provided as a Source Data file.

medium, the doubling time of the wild-type strain (WT) and the *meeF*, *meeY*, and FY mutants was comparable in liquid LB medium at 37 °C (Supplementary Fig. 1b). The Mn concentration in LB medium (<0.2 μM) is sufficient to support normal growth[16], but is far below toxic levels (~200 μM) for wild-type cells[15,25]. Thus, the reduced fitness of the FY mutant is unlikely to be related to Mn detoxification.

In contrast with LB, which is a complex medium, the FY mutant does not show a significant growth defect on glucose-minimal medium (MM) agar plates (Fig. 1b). To determine the origins of the small colony phenotype on LB plates, we compared growth of WT and FY mutant strains on MM plates amended with the individual ingredients of LB medium. The FY mutant was most defective (~67% decrease in colony area) in accessing the nutrients supplied as a tryptone (Fig. 1b), an enzymatic digest of casein. Thus, we hypothesized that FY is defective in secretion of feeding proteases required to degrade peptides[26]. Over half of the ~50 mM of amino acids in LB medium are only detected after acid hydrolysis[27], consistent with the idea that they are present predominantly as oligopeptides.

*B. subtilis* is widely appreciated for secreting proteins during transition and stationary phases[28], including industrially relevant proteases[29]. By monitoring extracellular protease production on milk agar plates[30], we observed halo formation up to a $10^{-4}$ dilution for WT cells, but only up to $10^{-1}$ dilution for FY (Fig. 1c). *B. subtilis* encodes

seven major extracellular feeding proteases (AprE, NprB, NprE, Epr, Bpr, Vpr and Mpr) with ~95% of activity attributed to the serine protease subtilisin (AprE) and the major metalloprotease NprE[28]. We used zymography to visualize the decreased extracellular protease activity in the FY mutant supernatant (Fig. 1d). The reduction in proteases was not restricted to the three metalloproteases (NprB, NprE, and Mpr) and included a reduction in several bands corresponding to degradation products of the large Bpr protease (154 kDa), as defined in previous studies[31] (Supplementary Fig. 2). This overall reduction in protease activity led us to hypothesize that the FY mutant has a generalized defect in protein secretion.

## The FY mutant is defective in protein secretion

The FY mutant secreted ~30% less protein than WT after overnight growth in LB (Fig. 2a). This reduction is not due to slower growth since the FY strain grows as well as WT in shaking cultures (Supplementary Fig. 1b). Analysis by silver-stained SDS-PAGE revealed a general reduction in protein levels for FY compared with the WT supernatant (Fig. 2c). Since TerC proteins have been implicated in translocation of metals across membranes[14,15], we hypothesized that they might play a specific role in metalation of Mn-requiring, extracellular proteins. This hypothesis is supported by analysis of Mn levels in the spent medium of overnight grown cells. Our LB medium contains 126 nM Mn

(Supplementary Fig. 3c), which is close to the minimal level needed to support good cell growth in minimal medium[16]. The spent medium of WT cells had 22 nM residual Mn, a > 5-fold reduction relative to the starting growth medium (Fig. 2b), which suggests that most Mn in the growth medium was imported to support growth. Even more dramatically, the spent medium from the FY mutant had only ~3.2 nM residual Mn (Fig. 2b). Thus, the higher levels of Mn (~22 nM) detected in the spent medium of WT (and single *meeF* and *meeY* mutants) is likely associated with secreted metalloproteins. In contrast, these strains had little difference in the residual level of Fe or Zn detected in the spent medium (Supplementary Fig. 3).

To further evaluate the reduced secretion capacity of the FY mutant (Fig. 2c), we monitored the extracellular levels of two well-studied secretion substrates: the major extracellular protease subtilisin E (AprE) and α-amylase from *B. amyloliquefaciens* (AmyQ)[28,29,32]. We observed a > 2-fold decrease for both AprE-FLAG[33] and AmyQ-His[34] in the FY mutant, but not in the *meeF* and *meeY* single mutants (Fig. 2, e, Supplementary Fig. 4). These observations indicate that MeeF and MeeY have overlapping functions required for efficient protein secretion.

The high-level overproduction of secreted proteins such as AmyQ can lead to activation of the secretion stress response controlled by the CssRS two-component system[28]. CssS senses the accumulation of misfolded exoproteins to activate expression of the quality control proteases HtrA and HtrB[35,36]. FY mutants did not experience secretion stress, as monitored using the CssR-dependent P*htrA-lux* reporter (Supplementary Fig. 5), consistent with their reduced secretion capacity. However, the P*htrA-lux* reporter was still able to be induced upon overexpression of AmyQ (Supplementary Fig. 5), despite reduced AmyQ export (Fig. 2e). Indeed, the FY mutant had a slightly increased P*htrA-lux* induction relative to WT (Supplementary Fig. 5), similar to other secretion-deficient mutants[37].

## MeeF and MeeY interact with the general secretory pathway
Most membrane and secreted proteins are translocated by the general secretion pathway (Sec pathway) in *B. subtilis*[38]. To investigate how MeeF and MeeY might influence secretion efficiency we generated strains expressing functional C-terminal FLAG-tagged MeeF or MeeY proteins under their native expression signals. Expression of both proteins was increased in LB medium supplemented with 50 μM Mn, and MeeY-FLAG had elevated expression in a mutant lacking MeeF (Supplementary Fig. 6a). Next, we used co-immunoprecipitation and untargeted proteomics to identify proteins that co-purify with C-terminal FLAG-tagged MeeF or MeeY (Table 1, Supplementary Fig. 6b, c). Overall, we identified more putative interaction partners for MeeF compared to MeeY, although many proteins were shared (Table 1). We focused on those putative interactors that are membrane proteins, since both MeeF and MeeY are integral membrane proteins comprising 7 transmembrane segments. Strikingly, many of the co-purifying proteins (Table 1) are either components of the holo-translocon (SecDF, SecY, YrbF)[39], quality control proteases (FtsH, PrsA, HtpX)[40–42], or subunits of the $F_1F_o$ ATPase[43], which form a complex with the SecYEG translocon. Thus, we hypothesized that MeeF and MeeY function as part of the secretosome[44].

## FY mutants require the FtsH protease for viability
FtsH is an ATP-dependent metalloprotease that functions to degrade membrane proteins in response to a variety of stresses[20]. Studies in *E. coli* indicate that FtsH selectively targets membrane-associated proteins that are misfolded, including partially translocated proteins stalled during passage through the SecYEG translocon (translocon jamming)[45]. Indeed, FtsH can degrade the major translocase subunit SecY[46] and this activity can even lead to a lethal defect in protein secretion if not properly regulated[47]. Phenotypically, *ftsH* mutants were similar to FY mutants in that they displayed a small colony size on

plates (Supplementary Fig. 7a). They were also defective for secretion of proteases (Supplementary Fig. 7b)[48], despite a near normal growth rate in LB broth (Supplementary Fig. 1b).

We hypothesized that nascent metalloproteins might jam the SecYEG translocon in the FY mutant leading to the observed global impairment in protein secretion. Under this condition, the FtsH protein is predicted to play an important role in removing partially translocated proteins and in clearing jammed translocons from the membrane[44]. To test this idea, we attempted to construct an FY *ftsH* triple mutant strain. While it was possible to generate all three possible double mutants, the triple mutant was inviable and efforts to construct this strain by genetic transformation invariably led to congression (acquisition of a functional copy of one of the missing genes). Conversely, induction of FtsH helped rescue the poor growth phenotype of FY mutant cells (Supplementary Fig. 7a). This is consistent with models that posit a role of FtsH in removal of partially translocated proteins from stalled translocons[44,49]. These genetic interactions support the hypothesis that MeeF and MeeY act during the translocation of nascent metalloproteins, and their absence leads to translocon jamming.

## MeeF and MeeY support activity of Mn-dependent lipoteichoic acid synthases
LtaS is the major lipoteichoic acid (LTA) synthase in *B. subtilis*[50,51]. LtaS is an integral membrane protein with an extracytosolic globular domain that requires Mn for its activity[51]. One phenotype of an *ltaS* mutant is small colony size[52], similar to FY (Supplementary Fig. 7a). We therefore hypothesized that MeeF and MeeY may play a role in the loading of Mn into LtaS.

LtaS is a low abundance constituent of the membrane proteome as monitored using quantitative proteomics[53]. Therefore, to determine if MeeF and MeeY have a role in the activation of LtaS we used immunoblotting to monitor levels of the LtaS product, LTA[54]. LTA levels were similar in the WT strain and the *meeF* single mutant, but reduced in the *meeY* mutant, and greatly reduced in the FY double mutant (Fig. 3a). The specificity of the assay is apparent from analysis of the *ltaS* single mutant, which lacked the abundant ~10–15 kDa LTA polymers. Cells mutant for *ltaS* experience cell envelope stress due in part to dysregulation of autolysins[54]. As a result, *ltaS* mutants have elevated expression of the σM cell wall stress response[55], which leads to expression of LtaSa(YfnI)[56]. This stress-induced LtaSa enzyme produces longer LTA chains[51] that were absent in the *ltaS ltaSa* double mutant (Fig. 3a). These results indicate that either MeeF or MeeY could support LtaS function.

Since FY mutants were defective in LTA synthesis, we predicted that they would also activate the σM stress response. Consistent with our hypothesis, the FY double mutant (but not *meeF* or *meeY* single mutants) had elevated expression of a σM-dependent promoter (P*sigM-lux*)[57] (Fig. 3b). The level of activation of σM in the FY mutant was comparable to that seen in a strain lacking *ltaS*. Further, σM activation was even higher in an *ltaS* strain additionally lacking *meeF (ltaS meeF)*, but not in the *ltaS meeY* double mutant (Fig. 3b). This additivity suggests that cell stress was increased by mutation of MeeF in strains lacking the major LtaS enzyme. Thus, LtaSa may also require Mn. Indeed, prior results demonstrate that an *ltaS ltaSa* double mutant has an elevated stress response[55], as also seen here (Fig. 3b). In addition to LtaS and the stress-induced synthase LtaSa, *B. subtilis* expresses YqgS (a minor LTA synthase) and YvgJ (an LTA primase)[51]. We monitored the expression of all four LTA synthesis genes using quantitative RT-PCR. As expected, the level of the σM-activated *ltaSa* mRNA was elevated in the FY mutant, but not in the single mutant strains (Supplementary Fig. 8).

Since the catalytic domain of LtaS is external to the cell, we next tested whether addition of Mn to the growth medium could activate enzyme that had been properly inserted in the membrane but had

**Table 1 | Membrane Proteins that Co-immunoprecipitate with MeeF and MeeY[a]**

| MeeF-FLAG (heating)[b] | MeeY-FLAG (heating)[b] | MeeY-FLAG (pH)[c] | Functional annotation |
|---|---|---|---|
| | | | *Holotranslocon Proteins* |
| SecY | | | Subunit of the SecYEG preprotein translocase |
| SecDF | SecDF | SecDF | PMF-dependent holotranslocon subunit |
| YrbF | | | Binds to SecDF as part of the holotranslocon (*E. coli* YajC ortholog; 34% identity) |
| | | | *Secretosome and Secretion-related Functions* |
| PrsA | PrsA | PrsA | post-translocation molecular chaperone |
| HtpX | | | Quality control membrane protease |
| FtsH | FtsH | | Quality control membrane protease |
| AtpA | AtpA | AtpA | ATP synthase (subunit alpha); intact ATP synthase interacts with SecYEG translocon[43] |
| AtpG | AtpG | | ATP synthase (gamma subunit) |
| | AtpF | | ATP synthase, part of the Fo complex |
| | | AtpD | ATP synthase, part of the F1 complex (subunit beta) |
| | | | *Other Membrane Proteins* |
| FloT | | | Flotillin, membrane-associated scaffold protein |
| FloA | FloA | | Flotillin, membrane-associated scaffold protein |
| MalA | | | 6-phospho-alpha-glucosidase |
| GlcD | | | possible glycolate oxidase subunit |
| NupN | | | lipoprotein, part of guanosine transporter |
| SwrC | SwrC | | resistance-nodulation-cell division (RND) |
| SsdC | | | spore shape determinant C (mother cell) |
| OppF | OppF | | Oligopeptide ABC transporter |
| OppB | | | Oligopeptide ABC transporter |
| FrlO | | | Aminosugar ABC transporter |
| SdhA | SdhA | | succinate dehydrogenase (flavoprotein subunit) |
| FhuD | | | hydroxamate siderophore ABC transporter |
| | QoxA | | cytochrome aa3 quinol oxidase (subunit II) |
| QoxB | | | cytochrome aa3 quinol oxidase (subunit I) |
| MsmX | | | multiple sugar ABC transporter (ATP-binding protein) |
| FtsA | | | cell division protein, member of the divisome |
| YknW | YknW | | modulator of ABC transporter assembly, SdpC secretion |
| | | MgtE | primary magnesium transporter |

[a]Only integral membrane proteins are listed here. The interacting proteins for MeeF and MeeY include 12 of the 40 most abundant membrane proteins, as measured previously[53], consistent with a physiologically relevant association with the secretosome complex.
[b]Immunoprecipitated fractions were eluted by heating, samples were treated with 1% TritonX-100 and their western analysis is shown in Supplementary Fig. 6b.
[c]Immunoprecipitated fractions were eluted by glycine (pH 3). The MeeY-FLAG sample is shown in Supplementary Fig. 6c.

failed to acquire its catalytic Mn ion. Indeed, amendment of LB medium with Mn restored LTA synthesis even in the FY mutant (Fig. 3a, inset). Further, Mn reversed the $\sigma^M$ cell envelope stress response in the FY mutant but not, as expected, in the *ltaS* mutant (Fig. 3b). Addition of Ca, which is not an effective co-factor for LtaS enzymes[51], did not reverse induction of the $\sigma^M$ stress response. This suggests that the FY mutant had properly expressed LtaS and inserted it in the membrane, but it likely failed to acquire Mn. Addition of Mn also partially reversed the $\sigma^M$ stress response seen in the *ltaS meeF* mutant (and *ltaS FY* mutant), consistent with the hypothesis that this strain may be partially deficient in activation of LtaSa or other back-up synthases. The elevated stress response seen in *ltaS ltaSa* (Fig. 3b) was also partially reduced by Mn, possibly due to increased activity of the YqgS synthase.

**FY mutants are sensitized to chemical inhibition of LtaS activity**
Because of its important role in physiology, LTA has been a target for the development new antibacterials[58]. An inhibitory compound LtaS-IN-1(1771) binds to the active site of LtaS[59,60] and inhibits the binding of its substrate phosphatidylglycerol[58]. We hypothesized

that mutants defective for metalation of LtaS might have altered sensitivity to the 1771 inhibitor. Indeed, the FY mutant was much more sensitive to growth inhibition by 3 μM 1771 than WT in LB broth (Fig. 3c). Further, *ltaS* mutant strains were as sensitive to growth inhibition by 3 μM 1771 as WT strain, presumably because 1771 is active against the alternative LtaS enzymes (Fig. 3c). We, therefore, hypothesized that the FY mutant was more vulnerable to 1771 because it was deficient in Mn loading in the active sites of LtaS and LtaSa enzymes. To explore this hypothesis, we tested the effect of metal ion supplementation on sensitivity to 1771 in various mutant strains. In the presence of 1771, Mn improved the growth of WT and FY mutant cells, but only slightly rescued the *ltaS* mutant (Supplementary Fig. 9). In contrast, Zn worsened the 1771 growth inhibition in all tested strains, in some cases dramatically (Supplementary Fig. 9). These results suggest that LTA synthases may be subject to mismetalation by Zn, and this inhibition was enhanced in the FY mutant where Mn acquisition was compromised. Collectively, these results indicate that 1771 likely binds to the active site of LtaS[60] and its paralogs, and this may hinder or preclude proper metalation under Mn-limiting conditions.

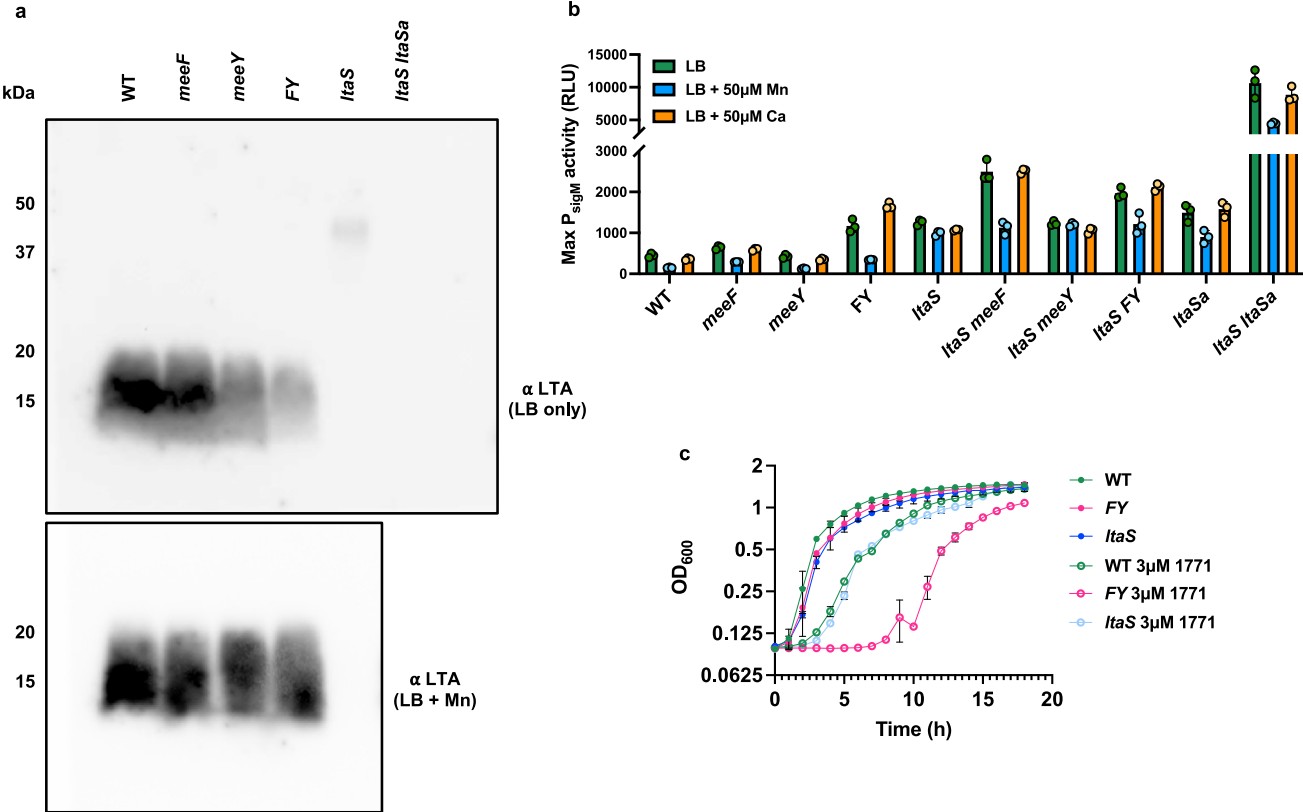

**Fig. 3 | FY mutants are defective in LTA synthesis. a** Immunoblot detection of LTA with anti-LTA monoclonal antibodies. Note that in *ltaS* mutants the signal in the -15–20 kDa range is absent, and instead longer polymers are detected that depend on the LtaSa enzyme[51]. The lower inset shows an immunoblot for cells grown in LB + 50 μM Mn. The images are representative of two independent experiments with gels loaded with extracts from equal cell numbers as judged by OD$_{600}$. Protein molecular weight markers are indicated on left under kDa. **b** Defects in LTA synthesis activate the σ$^M$-dependent cell envelope stress response as monitored using a luciferase transcriptional reporter fusion (P$_{sigM-luxABCD}$). Cells were grown in LB broth with or without Mn (50 μM) or Ca (50 μM). Data are from three independent experiments and shown with mean ± standard deviation, *n* = 3. **c** Defective activation of LTA synthase enzymes is associated with increased sensitivity to compound 1771, an LtaS inhibitor[58,105]. Aerobic growth of different strains (WT, FY, *ltaS*) in LB broth with or without 3 μM 1771 is shown. Data are representative of three independent experiments and are presented as mean ± standard deviation with the sample number *n* = 2. Additional results, showing the effects of metal supplementation are in Supplementary Fig. 9. Source data are provided as a Source Data file.

## The function of TerC proteins is conserved in gram-positive bacteria

TerC proteins are conserved among different bacterial species including several important pathogens[61]. Next, we tested the ability of selected heterologous TerC proteins (Fig. 4a) to restore fitness (colony size) and protease secretion to the FY mutant strain. As expected, expression of either of the native TerC proteins (MeeF or MeeY) restored colony size on LB. Similar results were seen with expression of two *Listeria monocytogenes* TerC proteins (Lmo0991, Lmo0992) and one from *B. anthracis* TerC (BanTerC) (Fig. 4b). Consistently, the protease activity of these complemented strains was also restored (Fig. 4c). Therefore, the function of TerC proteins is conserved across Gram-positive microbes.

## Discussion

The metabolic processes that support life would halt without the catalytic enhancements enabled by metal ions[2]. However, binding of the wrong metal (mismetalation) may lead to enzyme inactivation[62]. To ensure proper metalation, some Cu enzymes require metallochaperones[63,64], as do selected Zn enzymes in bacterial[65,66], fungal[67], and mammalian cells[68]. Acquisition of the proper metal ion presents a particular challenge for those membrane-associated and exported enzymes (exoenzymes) with active sites located outside the cytosol where metal concentrations are more variable. For high-affinity metals from the Irving-Williams series, acquisition from the environment may suffice to ensure metalation. For example, this is the likely route for Zn acquisition during the maturation of Zn-dependent metallo-beta-lactamases[11].

For lower affinity metals (including Mn and Fe), alternative strategies may be needed to ensure the proper metalation of exoenzymes. One common strategy is to assemble the functional metalloprotein within the cytosol, where metal concentrations are tightly regulated, and then to export the folded protein through the TAT secretion system. This is the strategy used for a Mn-dependent periplasmic cupin A (MncA) from *Synechocystis* PCC 6803, which acquires Mn inside the cell prior to export[10]. Biochemical studies reveal that MncA has a > 10$^4$-fold preference for binding to divalent Zn or Cu, but once metalated the bound Mn is kinetically trapped within the folded protein, which is therefore resistant to mismetalation[10]. *B. subtilis* also relies on TAT-dependent secretion for several metalloenzymes including a heme-containing peroxidase (EfeB), an iron-sulfur containing oxidoreductase (QcrA), and a Mn/Zn-dependent phosphatase (YkuE)[69]. Here, we provide evidence that TerC family proteins participate in an alternative pathway for the metalation of exoenzymes.

TerC proteins (Pfam03741) are a subgroup of the lysine exporter (LysE) superfamily of transporters and have seven TM segments and a conserved metal-binding site[14,70]. Although originally implicated in resistance to toxic tellurite salts[61], there is no evidence that TerC proteins transport tellurite[13]. We identified up-regulation of *meeF* in a screen for suppressors of the high Mn sensitivity of strains lacking the MneP and MneS Mn efflux proteins[16], and found that both MeeF and MeeY have Mn efflux activity[15]. The three *B. subtilis* TerC proteins

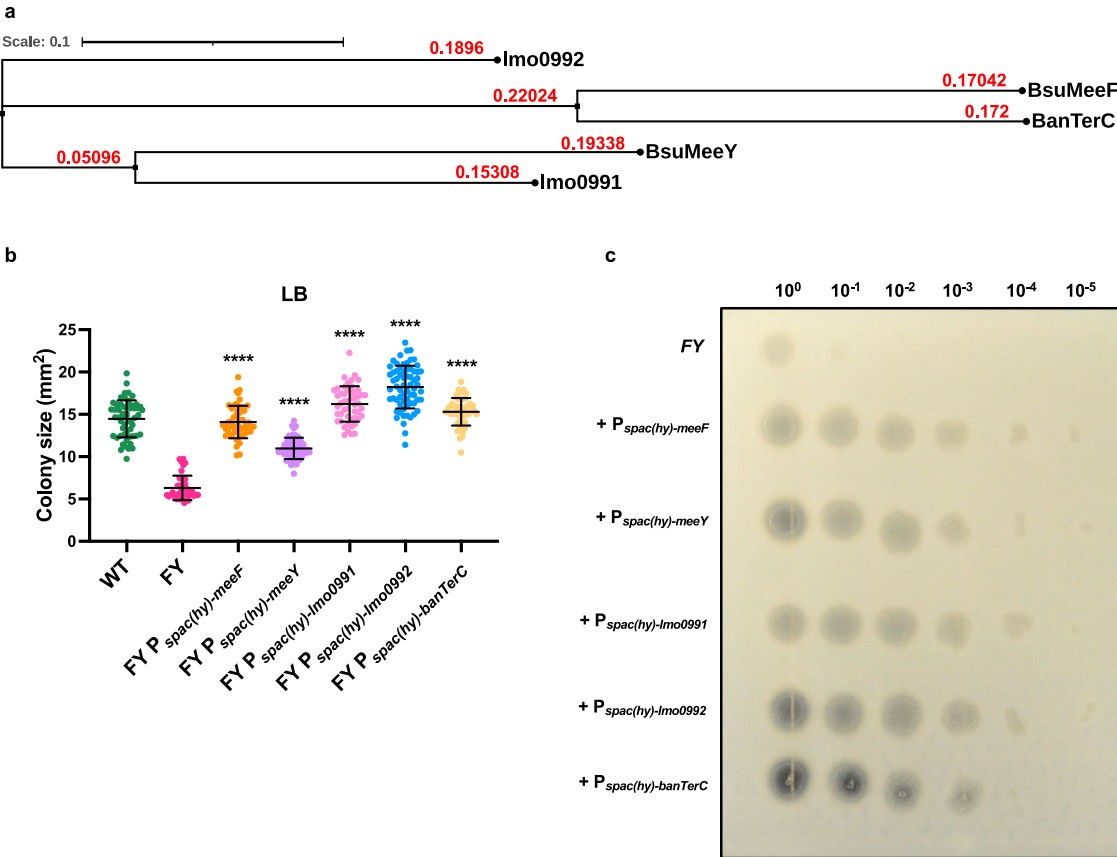

**Fig. 4 | Complementation of the FY mutant with orthologous TerC proteins.**
**a** Phylogenetic tree (branch lengths in red) comparing TerC proteins from *B. subtilis* (BsuMeeF, BsuMeeY) with homologs from *Listeria monocytogenes* (lmo0991, lmo0992) and *B. anthracis* (BanTerC). Protein sequences were aligned by MUSCLE (MUltiple Sequence Comparison by Log-Expectation)[106] using online analysis tools from EMBL-EBI[107]. The Newick data was then manipulated by Interactive Tree Of Life (iTOL) to display phylogenetic tree[108]. Scale bar = 0.1. Leaf nodes are shown as black circles and internal nodes are displayed as black squares (**b**) Colony sizes of FY mutant with induction of TerC proteins on LB medium with 50 μM IPTG as measured by imageJ. Isolated colonies from three independent cultures were measured for each strain and data are presented as mean ± standard deviation. *P* value of each strain compared to FY samples was calculated using Welch's *t* test, two-tailed, ****$p < 0.0001$. Sample size for each strain: WT, $n = 55$; FY, $n = 40$; FY $P_{spac(hy)-meeF}$, $n = 49$; FY $P_{spac(hy)-meeY}$, $n = 56$; FY $P_{spac(hy)-lmo0991}$, $n = 50$; FY $P_{spac(hy)-lmo0992}$, $n = 66$; FY $P_{spac(hy)-banTerC}$, $n = 48$. **c** Protease activities of FY mutant strains with expression of TerC proteins as measured on 5% milk agar plates. Cells were grown in LB broth with 50 μM IPTG inducer to $OD_{600}$ 0.4 and 2 μl of serially diluted cells ($10^0$–$10^{-5}$) were inoculated on the plates followed by incubation at 37 °C for 24 h. The image is representative of three independent experiments. Source data are provided as a Source Data file.

(MeeF, MeeY, and YjbE) are differentially regulated: *meeY* gene is regulated by a Mn-responsive riboswitch[71–73], *meeF* is part of the constitutively expressed *yceCDEmeeF* operon and can be further induced by stress-responsive sigma factors[56,74,75], and *yjbE* is most highly expressed during sporulation[24]. While the MeeF and MeeY proteins play overlapping roles in support of protease production (Figs. 1, 2) and activation of LtaS (Fig. 3), there may be other client proteins that are preferentially metalated by a specific TerC homolog, perhaps due to interactions with the variable, external loops of each homolog. Transcriptome studies reveal that *meeF* is the most highly expressed homolog over a range of growth conditions[24], *meeY* is optimally expressed in response to elevated Mn[71,72], and both *meeY* and *yjbE* are expressed during sporulation[24]. Thus, MeeF may serve to preferentially metalate those proteins most critical to growth, whereas MeeY and YjbE may additionally target proteins that are metalated when Mn is not a limiting resource and during sporulation. The role of TerC proteins in exoprotein metalation is likely important for bacterial pathogens that often face metal limitation due to host-imposed nutritional immunity[19,76]. For example, the neutrophil-released calprotectin protein sequesters metal ions and can thereby impose Mn limitation during infection[77].

Our findings support a model in which TerC proteins function as metallochaperones to load Mn into at least some nascent metalloproteins during their translocation through the secretion pathway. Most secreted proteins transit the membrane through the heterotrimeric SecYEG translocon driven by the SecA ATPase[38]. In *E. coli*, a larger complex, the holotranslocon, comprises SecYEG together with the SecDF-YajC accessory proteins and the YidC membrane protein insertase[39]. Further association with a variable set of folding chaperones and quality control proteases defines a larger secretosome complex[44]. Co-immunoprecipitation studies reveal an association of TerC proteins with the holotranslocon and secretosome[44] (Table 1). Interactions between TerC and holotranslocon proteins have also been reported in *E. coli* based on proteomic analyses of complexes enriched using co-immunoprecipitation and blue native polyacrylamide gel electrophoresis[78]. These findings support a role for TerC proteins during co-translocational metalation of nascent metalloproteins.

Strains lacking both MeeF and MeeY (FY mutants) are defective in protein secretion (Figs. 1,2), have a greatly reduced level of Mn in the cell supernatant (Fig. 2b), and are defective in LTA synthesis (Fig. 3). An inability to appropriately metalate nascent metalloproteins may contribute to translocon-jamming. Jamming of the translocon in the FY mutant is consistent with the similar phenotypes noted for the FY and *ftsH* mutant strains, both of which are slow growing on plates and have a reduced capacity for protein secretion (Supplementary Fig. 7). FtsH serves as a quality control protease that helps rescue jammed

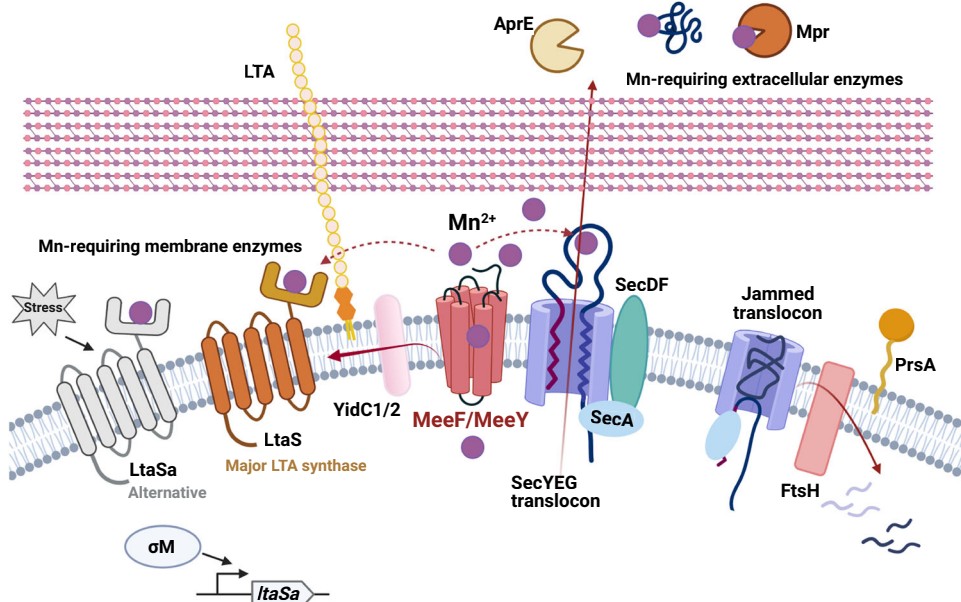

**Fig. 5 | The functions of TerC proteins MeeF and MeeY in exoenzyme metalation.** MeeF and MeeY are integral membrane proteins that function in Mn export[15]. MeeF and MeeY are here shown exporting Mn ion (purple circles) to support metalation of exoenzymes. MeeF and MeeY interact physically (co-immunoprecipitation) and genetically (epistasis with *ftsH*) with proteins of the secretosome. These results suggest that MeeF and MeeY function co-translocationally to insert Mn into nascent metalloproteins. As a result, *meeF meeY* (FY) double mutants are deficient in Sec-dependent secretion of exoenzymes (e.g., proteases, AprE, AmyQ), which leads to growth defects on LB medium. FY mutants are also deficient in activation of LTA synthases, which bind Mn to an extracellular catalytic domain. The essentiality of FtsH in the FY mutant is consistent with jamming of the SecYEG translocon. MeeF and MeeY may function as metallochaperones that directly transfer Mn to client proteins, and they may help generate a sufficiently high local Mn concentration to allow metalation. Created with BioRender.com.

translocons, likely by degradation of partially translocated proteins[49], and this activity is essential in the FY mutant.

In addition to transfer of Mn to nascent metalloproteins during translocation, TerC proteins may also function by export of Mn to generate a sufficiently high local concentration to ensure metalation of Mn-requiring enzymes (Fig. 5). Mn and other divalent cations are concentrated near the cell surface by association with the negatively charged cell wall polymers[79,80]. Teichoic acids, in particular, have a high ion-binding capacity which supports both metal import and protein folding[81,82]. Future studies will be needed to establish how weak-binding metals such as Mn are retained by exoenzymes. For some proteins, Mn may be oxidized to less exchangeable Mn(III), or the bound Mn may be kinetically trapped after protein folding. Alternatively, MeeF and MeeY may be able to repeatedly load metal into those proteins that are retained in the membrane (LtaS) or in the vicinity.

Our findings suggest that bacterial TerC proteins function, at least in part, by mediating the co-translocational insertion of Mn into nascent proteins during transit across membranes. This pathway provides an alternative to post-translocational metalation, which can be facilitated by periplasmic metallochaperones. While TAT-dependent secretion is also used for export of metalloenzymes, this pathway is not well suited for membrane-embedded enzymes or those that may be toxic if activated in the cytosol. *B. subtilis* is an important biotechnology platform often employed to produce secreted proteins[83]. Protein secretion in this organism has been extensively studied[84] with a focus on optimization of protein yields[37,85,86]. The involvement of MeeF and MeeY as potential members of the secretosome complex suggests a previously unappreciated feature of protein secretion in this system.

Functional studies have linked diverse TerC and related UPF0016 proteins to the transport of Mn and Ca[14,19,87,88]. In plants, *Arabidopsis thaliana* (AtTerC) is important for the insertion of thylakoid membrane proteins[21,89]. AtTerC interacts with the membrane protein insertase ALB3, a YidC ortholog[21], consistent with a function in co-translocational protein metalation in support of photosystem II

assembly[90]. In yeast, Gdt1p mediates Mn influx into the Golgi to activate metalloenzymes functioning in protein glycosylation[91–93]. The human ortholog TMEM165 has a similar role as Gdt1p and missense mutations are associated with congenital disorders of glycosylation (CDG)[91,94–96]. These transporters all help ensure that enzymes exported from the cytosol or embedded in membranes are appropriately metalated. Further studies of the detailed mechanisms of bacterial TerC proteins, and their role in supporting metalloprotein function, may provide insights relevant to the related metalation pathways in diverse organisms.

## Methods
### Bacterial strains and growth conditions
All strains used in this study are listed in Supplementary Table 1. Mutant strains were obtained from the *Bacillus* Genetic Stock Center (BGSC) as erythromycin marked gene disruptants from the BKE collection[97]. Mutations were transformed into the desired strain and markerless in-frame mutants were generated by transformation with plasmid pDR244 to remove the erythromycin cassette[97]. Gene deletions were confirmed by PCR analysis using flanking or internal primers (Supplementary Table 1). The AmyQ-His overexpression plasmid pKTH10[34] was selected with 15 μg/ml kanamycin.

For construction of FLAG-tagged gene fusions we PCR amplified the C-terminal ~500–700 bp of the *meeF, meeY* and *aprE* genes with primers (Supplementary Table 1). The PCR products were restriction digested and ligated into pre-digested pMUTIN-FLAG[98] using T4 DNA ligase (NEB). The constructs were transformed into *E. coli* DH5α and TG1 strains selected with ampicillin (100 μg/ml). The recombinant plasmids were transformed into *B. subtilis* and integrated into the chromosome with erythromycin (1 μg/ml) selection (Supplementary Table 1). For IPTG-based $P_{spac(hy)}$ plasmids, genes were amplified using PCR with high-fidelity Phusion polymerase (NEB) and the PCR products were digested with restriction enzymes (XbaI and BglII) and ligated into pPL82[99]. The ligation products were transformed into *E. coli* DH5α. Plasmid constructs were verified and integrated at the *amyE* locus by

transformation of recombinant plasmids into cells using modified competence (MC) media.

## Growth conditions

Bacteria were grown in liquid or on solid lysogeny broth (LB) (Affymetrix) at 37 °C unless otherwise stated. LB medium contains 10 g tryptone, 5 g yeast extract, and 5 g NaCl per liter. Antibiotics used for selecting *B. subtilis* strains include: spectinomycin 100 µg/ml, macrolide-lincosamide-streptogramin B (MLS = 1 µg/ml erythromycin +25 µg/ml lincomycin), kanamycin 15 µg/ml, and chloramphenicol 10 µg/ml. Glucose-minimal medium (MM) was prepared as a 2X MM stock made using a 10X Bacillus salt solution [$(NH_4)_2SO_4$ 20 g/L, $Na_3C_6H_5O_7\cdot2H_2O$ 10 g/L, L-glutamic acid potassium salt monohydrate 10 g/L], and adding 80 mM MOPS buffer (pH 7.4 using KOH), 4 mM $KPO_4$ (pH 7.0), 20 µg/L tryptophan, 2% glucose, 160 µM $MnCl_2$, 1.6 mM $MgSO_4$, 8.8 mg/L ferric ammonium citrate. For pouring plates, equal volumes of filter-sterilized 2X MM stock and 3% autoclaved agar were mixed. 1% Tryptone, 1% NaCl or 0.5% Yeast Extract (Y.E.) were added into MM where indicated.

## Colony size measurements

Bacteria were grown in LB broth or liquid minimal medium at 37 °C with vigorous shaking to mid-exponential phase ($OD_{600}$ ~ 0.4–0.5), serially diluted, and 100 µl of different dilutions was plated onto 15 ml fresh LB (dilution $10^{-5}$ to $10^{-6}$) or MM agar plates (dilution $10^{-1}$ to $10^{-5}$) with different amendments as noted to get isolated colonies. Plates were incubated at 37 °C for 24 h prior to imaging. Colony size was measured using Fiji ImageJ[100].

## Protease activity on skim milk agar

Casein degradation in skimmed milk agar plates (5% skim milk and 1% agar) was used to assess protease activity by formation of a clear zone. Bacterial cells were grown in LB broth at 37 °C with vigorous shaking to mid-exponential phase ($OD_{600}$ ~ 0.4). Cultures of identical OD were serially diluted from $10^0$ to $10^{-5}$. 2 µl of cells were inoculated on the plates. Plates were incubated at 37 °C for 24 h and then imaged.

## Proteolytic profile by zymography

Zymography was performed as described previously[101] and proteases assigned as described[31]. The thickness of the zymography gel was 1.5 mm, and the resolving gel contained 10% gelatin. Supernatants from 1 ml of overnight grown cultures were centrifuged at 15,000 *g* for 10 min, and then mixed with 2X sample buffer without reducing agent incubated at 37 °C for 30 min. 12 µl of each sample was then loaded on an SDS-PAGE gel. After electrophoresis, the gels were placed in the renaturing buffer (2.5% Triton X-100) and incubated at room temperature for 30 min without shaking. The gel was washed twice with water for 5 min and then placed into activation buffer (50 mM Tris-HCl, pH 7.5, 1% Triton X-100 and 25 mM $CaCl_2$) and incubated at 37 °C for 18 h. The gels were stained by Coomassie blue for 2 h and destained overnight (10% acetic acid, 40% methanol) prior to imaging. Note that this assay is selective for those proteases that are easily renatured following SDS-PAGE and have activity with gelatin[101].

## Metal ion quantification by ICP-MS

Metal content of supernatant fractions was measured as described previously[102]. Cells were grown overnight in LB broth, and the supernatants obtained by centrifuging 2 ml of the cultures at 15,000 *g* for 10 min. The protein concentrations of the supernatants were measured using the Bradford assay (Bio-Rad, USA). Following protein estimation, 900 µl of supernatant samples were mixed with 600 µl of buffer (5% $HNO_3$, 0.1% Triton X-100) and incubated at 95 °C for 30 min. After centrifuging the samples at 15,000 *g* for 10 min, 1 ml of clear supernatant fraction was transferred to new tubes, and the total metal

ions were analyzed using a Perkin-Elmer Elan DRC II ICP-MS as described[102].

## Silver stain for protein detection in polyacrylamide gels

Cells were grown aerobically overnight in LB broth at 37 °C. Supernatants were obtained by spinning down 2 ml of the cultures at 15,000 *g* for 10 min and filtered using a low protein-binding polyethersulfone (PES) membrane sterile filter (Foxx Life Sciences). 500 µl of the supernatant was then mixed with 2X Laemmli sample buffer (Bio-Rad, USA) and were boiled at 95 °C for 5 min. The samples were centrifuged and 10 µl of each supernatant was loaded onto a 4–20% stain-free polyacrylamide gel (Bio-Rad, USA). After electrophoresis, the gel was stained as per the recommendations of the Pierce silver stain kit (Thermo Scientific™, catalog #24600). Briefly, the gel was fixed in fixing solution (30% ethanol, 10% acetic acid). After ethanol and water wash, the gel was incubated in sensitizer working solution and then stained. After adding developer, the bands appear on the gel after 2-3 min. Stop solution (5% acetic acid) was used to terminate gel development and the gel was imaged using a GelDoc Gel imaging system (Bio-Rad, USA).

## Protein detection by Western blot

Samples were collected from overnight cultures without or with 50 µM Mn. About 5 ml of cultures of identical OD were centrifuged and pellets and supernatants were collected. Cells were resuspended in 100 µl lysis buffer (20 mM Tris-HCl pH8.0, 1 mM EDTA, 1 mg/ml lysozyme) at 37 °C for 30 min. The crude cell lysates and supernatant fractions were mixed with 2X Laemmli sample buffer (Bio-Rad, USA) and boiled at 95 °C for 10 min. After centrifugation, 12 µl was electrophoresed on a 4–20% strain-free polyacrylamide gel (Bio-Rad). Proteins were transferred onto a PVDF membrane using a Trans-Blot Turbo Transfer System (Bio-Rad, USA). The PVDF membrane was stained with Ponceau S dye (5% glacial acetic acid, 0.1% ponceau S tetrasodium salt) for 15 min. The image was taken after removing non-specifically bound Ponceau S by rinsing the membrane with TTBS (1X TBS with 0.1% Triton X-100). The membrane was blocked with 5% protein blotting blocker dissolved in TTBS for 1 h and incubated overnight at room temperature with the primary antibodies (1:5000), rabbit anti-FLAG antibody (Sigma, catalog #F7425-.2MG) or mouse anti-6X-His Tag antibody (Invitrogen, catalog #MA1-21315). After washing the membrane with 1X TTBS 4 times, the membrane was incubated with the secondary antibodies (1:10000), goat anti-rabbit IgG (H + L), (HRP-linked, Invitrogen, catalog #65–6120) or rabbit anti-mouse IgG (H + L), (HRP-linked, Invitrogen, catalog #61–6520). The membrane was washed five times with 1X TTBS and then developed using the Clarity Western ECL substrate (Bio-Rad, USA) and subsequently documented using GelDoc Gel imaging system (Bio-Rad, USA). The band intensities were quantified using ImageJ software.

## LTA detection by Western blot

Samples were collected as described[54]. Briefly, strains were grown aerobically in 5 ml of nutrient broth (NB) with 5 mM $MgSO_4$ at 30 °C overnight. Cultures were diluted to 0.01 $OD_{600}$ in NB with or without 50 µM Mn, and then grown with shaking at 37 °C for about 5 h. Cultures with similar cell numbers ($OD_{600}$ ~ 0.6) were collected. Cells were centrifuged at 15,000 *g* for 10 min and then washed by 100 µl of solution A (10 ml 100 mM Tris-HCl [pH 7.4] with one cOmplete Mini EDTA-free protease inhibitor cocktail tablet, Roche) as described[54]. Cells were resuspended in 35 µl of solution A and 35 µl of 2X Laemmli sample buffer without reducing agent, and incubated at 100 °C for 45 min. Samples were on ice for 3 min then mixed with 10 U/ml DNase at 37 °C for 30 min. After centrifuging, supernatants were collected and storage at −20 °C. For analysis, samples with 48 µl of supernatant and 2 µl of beta-mercaptoethanol were heated at 95 °C for 10 min. After quick

centrifugation, 12 µl was electrophoresed on two 4–20% strain-free polyacrylamide gels (Bio-Rad). One gel was stained by commassie blue as a loading control and the other was transferred onto a PVDF membrane using a Trans-Blot Turbo Transfer System (Bio-Rad, USA). The transferred membrane was incubated with shaking in the 1X PBS buffer with 3% bovine serum albumin (BSA). LTA synthesis was monitored by Western blot analysis by incubation for 1 h and 30 min with Gram-positive LTA monoclonal antibody (Thermo Fisher, catalogue #MA1-7402) at a dilution of 1:1,000 in 1X PBS buffer with 3% BSA. The membrane was then washed prior to incubation for 1 h with the secondary antibody, rabbit anti-mouse IgG (H + L), (HRP-linked, 1:10000, Invitrogen, catalog #61-6520) in 5% protein blotting blocker dissolved in 1X PBS. The membrane was developed and imaged as above using the Clarity Western ECL substrate (Bio-Rad, USA).

### Co-immunoprecipitation (Co-IP) of MeeF-FLAG and MeeY-FLAG

Cells were collected from 10 ml of overnight LB cultures grown with 50 µM Mn and lysed with 1 ml of lysis buffer (Tris-HCl, 50 mM EDTA, and 1 mg/ml lysozyme) at 37 °C for 30 min. The samples were sonicated for 5 min and incubated with 10 U/ml DNase at 37 °C for 30 min. 1% Triton X-100 was added, and then samples were incubated on ice for 2 h before adding anti-FLAG M2 magnetic beads (Millipore Sigma). Co-IP samples were prepared as per manufacturer's instructions. Samples were end-over-end rotated with anti-FLAG M2 magnetic beads at 30 °C for 3 h and were then placed on a magnetic stand for a pull-down. Supernatants were discarded and the samples were washed with 1X PBS buffer three times by end-over-end rotation for 30 min. Samples were eluted by either heating at 95 °C for 10 min with 50 µl of 2X Laemmli sample buffer (no reducing agent) or by treating with 50 µl glycine (pH 3.0) for 30 min. Sample supernatants were separated from the magnetic beads by placing in a magnetic stand and transferred into new tubes. Eluted proteins were electrophoresed and immunodetected to verify recovery of the FLAG-tagged protein. One MeeF-FLAG sample, two independent MeeY-FLAG samples, and two control samples (WT without FLAG tag) were sent for identification of recovered proteins by trypsin digestion followed by peptide analysis using liquid chromatography with tandem mass spectrometry (LC-MS/MS) at the Cornell Proteomics and Metabolomics Facility. The MS raw files were searched by Proteome Discoverer 2.5 against the *Bacillus subtilis* NCBI database, which has 197740 sequences, along with a common contaminant database (246 entries) with 2-missed cleavages of trypsin allowed. Oxidation of M, deamidation of N and Q were specified as dynamic modifications; protein N-terminal acetylation, M-loss, and M-loss+acetylation was set as a variable modification; carbamidomethyl C was specified as a static modification. The initial search results were filtered with 5 ppm for each peptide and protein hits in the contaminant database were removed. Only proteins with at least 2 peptides were considered as positive identifications. Membrane proteins identified in the experimental samples but not in the controls were sorted in Table 1.

### Promoter-luciferase measurements

The activity measurements of promoter-*lux* reporter fusions for $P_{htrA}$ and $P_{sigM}$ were conducted as described previously[103]. Cells were grown aerobically in LB broth at 37 °C to $OD_{600}$ ~ 0.4. 1 µl of the cultures were inoculated into 99 µl of fresh liquid LB without or with different metals dispensed in a 96-well plate. Aerobic growth ($OD_{600}$) and luminescence were measured every 6 min using a Synergy H1 (BioTek Instruments, Inc. VT) plate reader. The maximum promoter activity was measured after normalizing relative light units (RLU) to culture density.

### Real Time RT PCR

Total RNA was extracted from 1.5 ml of mid-log (OD 600 nm = 0.4) WT, *meeF*, *meeY*, and FY grown in LB broth using a QIAGEN kit. Total RNA was treated with DNase (Ambion) enzyme to further purify and remove traces of DNA. For each reaction 2 µg RNA was used for cDNA synthesis facilitated using High-Capacity reverse transcriptase (Applied Biosystems) amplified with random hexamer primers. Further, for amplicon measurements 10 ng of cDNA was used as a template along with 500 nM of *ltaS*, *ltaSa(yfnI)*, *yqgS*, *yvgJ*, and *gyrA* (control) gene specific qPCR F/R primers in a 1X SYBR green master mix (Bio-Rad). Threshold and baselines parameters were kept consistent for experiments performed on a different day during data analysis. All Ct mean values were normalized to *gyrA* ($n = 4$).

### Growth measurement with LtaS inhibitor 1771

Cultures were grown aerobically in LB broth at 37 °C to $OD_{600}$ ~ 0.4. In a 96-well plate, 2 µl of the inoculum was added to 198 µl of LB supplemented with different concentrations of 1771 (MedChemExpress) or different metals (50 µM Mn, 50 µM Zn). The change in growth at 600 nm was monitored as a function of time periodically using a Synergy H1 (BioTek Instruments, Inc. VT) plate reader.

### Reporting summary

Further information on research design is available in the Nature Portfolio Reporting Summary linked to this article.

## Data availability

The mass spectrometry proteomics data have been deposited to the ProteomeXchange Consortium via the PRIDE partner repository[104] with the dataset identifier PXD044879. All other data supporting the findings of this study are available within the paper and Supplementary Information or Source Data files. Source data are provided with this paper.

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

## Acknowledgements

We acknowledge Sri Paruthiyil for his initial observation of the small colony phenotype, Brian Wendel for help with ICP-MS operation, the Cornell Biotechnology Resource center for proteomics analysis, and Prof. Jan Maarten Van Dijl for the gift of pKTH10 plasmid. This work was supported by National Institutes of Health grant R35GM122461 awarded to JDH. The content is solely the responsibility of the authors and does not necessarily represent the official views of the National Institutes of Health.

## Author contributions

B.H., A.J.S. and J.D.H designed the experiments. B.H. and A.J.S. performed the experiments. B.H, A.J.S. and J.D.H wrote the paper.

## Competing interests

The authors declare no competing interests.
