## [Peer Review File · Nature Communications]

REVIEWER COMMENTS

Reviewer #1 (Remarks to the Author):

How the correct metals locate to the correct proteins is a puzzle. Moreover, this is an important puzzle since almost half the reactions of life are estimated to be metal catalysed and typically alternative metals will not drive catalysis. The puzzle is further complicated by the observation that most metallo-proteins bind one or more wrong metals many orders of magnitude more tightly than the required metal. For cytosolic proteins the puzzle is substantially resolved by the actions of metal homeostatic systems: These systems maintain the availabilities of the tightest binding metals much lower than the availabilities of the weaker binding ones and so it becomes possible to simultaneously have proteins that require tight binding metals and those that require weak binding ones in the same cytosolic compartment. This becomes more challenging however outside the plasma-membrane, especially of Gram positive bacteria, since this location is essentially continuous with the external environment. Intriguingly, in this manuscript we learn that even manganese-proteins (available divalent manganese being a relatively weak binding metal) that are secreted in an unfolded state, still take advantage of the homeostatic mechanisms that control cytosolic metal availabilities to the inverse of the Universal affinity series (the Irving Williams series). Previously, we only knew this to be true of manganese-proteins that are secreted in a folded state via the TAT system. In this manner metal specificity becomes the product of the multiple partitioning events that control cytosolic metal availabilities for both TAT and Sec substrates. This is an significant advance which adds to a body of literature that needs to be brought to the attention of the widest possible audience.

Do we know if manganese becomes kinetically trapped in Lipoteichoic acid synthase post-folding, or if the exchange of manganese with a tighter binding metal such as zinc is slow? Could this be experimentally addressed or alternatively could the authors add further comments on how, post MeeF/Y-assisted metalation and secretion, subsequent exchange of manganese for a tighter binding metal is avoided.

Is it known what fraction of the Sec secretome is composed of LTA? Which data exclude the possibility that MeeF/Y act directly in the secretion of LTA but only indirectly exert effects on manganese because LTA is a major sink for extra plasma-membrane manganese (perhaps bound in less-exchangeable oxidised forms)?

Could the inhibitory phenotypes related to zinc be a consequence of known inhibitory effects of zinc on the mechanisms of manganese import as documented by Chris McDevitt and co-workers?

line 251. Metalation of CucA was subsequently shown to require two copper transporting ATPases and a cytosolic metallochaperone, with CucA secretion being absent in mutants lacking these proteins suggesting that metalation is assisted and co-coincident with secretion (J Biol Chem 2010 285: 32504-32511).

The use of "suggest" and "possibly" in the closing sentence: 'These results "suggest" that TerC proteins are important in both bacteria and eukaryotes for the proper functioning of exported proteins, "possibly" by mediating the co-translocational insertion of metals into nascent proteins during transit across membranes' is more cautious than elsewhere and perhaps some statements need to be toned-down and/or caveats added.

Reviewer #2 (Remarks to the Author):

The manuscript by He and colleagues describes a novel role for TerC proteins in the delivery of manganese to proteins exported via the SEC translocase. The findings built on their own analyses which defined TerC as manganese export systems, and the authors introduce these proteins as possibly functionally conserved across multiple kingdoms of life. The work performed here is of a high standard and the manuscript is very clearly written. The experiments performed here are logical and provide the necessary information required to draw conclusions regarding the proposed novel role for TerC in *B. subtilis*. However, the brevity of the manuscript presents a limitation in terms of drawing

broader conclusions across bacteria and definitely when considering other kingdoms of life.

Main comments:

This work may benefit from some *in silico* work, including the genetic organisation and conservation across the species, genus, between other bacteria and across kingdoms. Structural models are likely to assist too. These don't need to be described in detail in the main body of the text, but may appeal to those that are not in the field of *Bacillus* research.

Expanding upon this, I would like to learn more about why MeeF appears to have a slightly stronger functional role than MeeY, and why more interacting proteins were identified. Other than the 40% similarity, what's different: expression, structure, distribution across the cell surface?

Why is YjbE not contributing to the processes described here, and how may it fulfill its role during sporulation? Please discuss.

How are TerC proteins and their role in metalation of secreted proteins impacted by Mn starvation, something that bacterial pathogens readily encounter during infection?

Minor comments:

Line 64: Please don't specify the name of the mutant in the introduction, leave that until the results, where it has already been clearly stated too.

Line 75: This statement is superfluous, as this can be derived from the intro.

Line 643: Is it clearer bands, or simply more bands?

Line 101: Listing the molecular weights of the proteins seems confusing and Fig 1d doesn't actually show the specifics of those proteases listed, only that of the delta7 strain. Hence, denoting specific candidates seems appropriate for Fig S2 alone.

Line 654: Please indicate total samples (were multiple biological replicates included on one day, as I assume that technical replicates are not displayed)

Line 212: Please use these references for the strains with FLAG tag, as it may cause confusion.

Line 138-139: This should be the other way around.

Line 803: This sentence seems incomplete.

Line 145: Please define that the MeeF interactome is broader than that of the MeeY.

Line 811-812: I would say that the pattern is similar, it's the difference in intensity that is more striking.

Lines 183-185: Bit odd to explain what they do after already defining the result, restructure this order.

Fig S8: Relative transcript to what? Scale is confusing. Why only two reps?

Line 209: Please ascertain this transcriptionally.

Lines 229-236: This really needs bioinformatics to support the choice of those two candidates and the claim that it's conserved.

Lines 239-246: This can be condensed.

Line 259: delete "its".

Line 266: Is Mn structural or any role in folding, as lack of metalation seems like an insignificant reason to not release the protein by the translocase? Or perhaps degradation is favourable as the protein would be non-functional, please discuss.

The legends of Fig 2D and S2 appear contradictory regarding the role of AprE.

Reviewer #3 (Remarks to the Author):

This MS addresses the important issue of the metalation of extracytoplasmic proteins. This is important because many secretory Gram-positive proteins are metalloproteins or require metal ions for folding following translocation across the membrane (via the Sec translocase) in an essential unfolded/unstructured form. Slowly folding proteins are subject to proteolysis by quality control proteases and metal ions are known to act as folding factors, speeding up the folding at least for some proteins. This MS provides a mechanism for the metalation by the lower affinity metal ion Mn, so as to potentially optimise folding and avoid mis-metalation by higher affinity metal ions. There is a suggestion that folding provides an energy force that helps "pull" for the protein from the translocase and so metalation may help in this process.

The current view is that metalation occurs in most cases via the concentration (of the medium) of metal ions in the negatively charged cell wall. However, this does not allow for mis-metalation for lower affinity metal ions. Therefore, the hypothesis proposed in this manuscript is attractive. I do, however, have some comments on what I see as potential experimental limitations that the authors should address. These are indicated below.

L104/Fig 1c/d

The legend states that "Supernatants were collected from ONCs with the same cell number(s)". What medium was used and did all the cultures reach stationary phase with the same numbers of cells and at the same time? I might have expected the FY mutant to have lower numbers or to have a slower increase in some media. I am also cautious about simply using data derived from overnight cultures (ONCs) without considering growth kinetics. *Bacillus* continues to produce many secretory enzymes in stationary phase and therefore cells that enter stationary phase earlier than others could generate more enzyme during their longer stationary phase. Therefore, the lower protease activity from the mutant could be due to a slower growth rate (ie reaching stationary phase later) and a resulting shorter time in stationary phase. Has this been taken into account?

L 107

The same point as above, although I suspect the interpretation is correct. The length of time in stationary phase could significantly affect secretory protein production and a better experiment would have been to take samples at a fixed time following the transition from exponential to stationary phase. The >5-fold reduction in extracellular Mn is to be expected from previous work on these genes as Mn exporters.

L117

As above, and AmyQ (which requires Ca ions for folding) certainly accumulates in stationary phase.

L127

It is well known that AmyQ induces secretion stress as compared with native enzymes. As a Ca requiring enzyme it is not clear what the significance of these findings are. I think the most significant finding here is that secretion stress is not induced in the FY mutant, which I find surprising if Mn requiring enzymes are unable to fold/fold rapidly.

L142

It could be argued that the results in Table 1 indicate that the MeeF and MeeY (membrane proteins themselves) simply interact with other membrane proteins. I don't see a clear justification, based on these data, for claiming MeeF and MeeY appear to function as part of the secretosome.

L156

Same point as above about ONCs and secretory proteins.

L168

Despite my comments about the way protease levels were estimated, the potential role of MeeF and MeeY in providing Mn as a folding factor helping to "pull" proteins from the translocase would be consistent with the role of FtsH clearing the translocase in the FY mutants.

L204

The data on Mn addition to the medium in relation to LtaS seem clear. However, the medium (as indicated above) already contains some Mn that, in the wild type, would be at a higher concentration at the membrane/wall interface due to its mobile interaction with the phosphate in LTA. Could the influence of Mn addition on LtaS activity in the FY mutant be simply the result of controlling the level of Mn added to the medium to "just enough" (as indicated above) and the influence on the WT as indicated below) for the wild type but not enough for the mutant. If LtaS folding was facilitated simply by the level of Mn in the medium, and this was lower in the FY mutant due to Mn retention by the cell, then the rate of folding of LtaS at lower Mn concentrations could be slower, leading to its removal by quality control proteases. Additional Mn allow this to recover LTA synthesis.

L264

From the data presented I think it is too early to argue that MeeF and MeeY are accessory subunits of the holotranslocon

RESPONSE TO REVIEWER COMMENTS

Reviewer #1 (Remarks to the Author):

How the correct metals locate to the correct proteins is a puzzle. Moreover, this is an important puzzle since almost half the reactions of life are estimated to be metal catalysed and typically alternative metals will not drive catalysis. The puzzle is further complicated by the observation that most metallo-proteins bind one or more wrong metals many orders of magnitude more tightly than the required metal. For cytosolic proteins the puzzle is substantially resolved by the actions of metal homeostatic systems: These systems maintain the availabilities of the tightest binding metals much lower than the availabilities of the weaker binding ones and so it becomes possible to simultaneously have proteins that require tight binding metals and those that require weak binding ones in the same cytosolic compartment. This becomes more challenging however outside the plasma-membrane, especially of Gram positive bacteria, since this location is essentially continuous with the external environment. Intriguingly, in this manuscript we learn that even manganese-proteins (available divalent manganese being a relatively weak binding metal) that are secreted in an unfolded state, still take advantage of the homeostatic mechanisms that control cytosolic metal availabilities to the inverse of the Universal affinity series (the Irving Williams series). Previously, we only knew this to be true of manganese-proteins that are secreted in a folded state via the TAT system. In this manner metal specificity becomes the product of the multiple partitioning events that control cytosolic metal availabilities for both TAT and Sec substrates. This is a significant advance which adds to a body of literature that needs to be brought to the attention of the widest possible audience.

**** We appreciate the detailed summary of the referee who has judged this work to be highly significant.**

Do we know if manganese becomes kinetically trapped in Lipoteichoic acid synthase post-folding, or if the exchange of manganese with a tighter binding metal such as zinc is slow? Could this be experimentally addressed or alternatively could the authors add further comments on how, post MeeF/Y-assisted metalation and secretion, subsequent exchange of manganese for a tighter binding metal is avoided.

**** We agree that the problem of mismetalation is important and that the Mn bound LtaS enzyme might, under some conditions, be inhibited by tighter binding metals. We address this experimentally by showing that mutants deficient in metalation of LtaS with Mn (FY mutants) are sensitive to inhibition by Zn, consistent with a mismetalation mechanism (Figure S9). Further studies of metal exchange after metal loading will require future study.**

Is it known what fraction of the Sec secretome is composed of LTA?

**** We believe the referee meant to inquire about how abundant the LtaS protein is within the Sec secretome. The secretome of *B. subtilis* includes both extracellular enzymes and proteins destined for the membrane (the membrane proteome). The text has been revised to clarify that LtaS is a low abundance membrane protein (line 194), as shown previously (PMID 31424929).**

Which data exclude the possibility that MeeF/Y act directly in the secretion of LTA but only indirectly exert effects on manganese because LTA is a major sink for extra plasma-membrane manganese (perhaps bound in less-exchangeable oxidised forms)?

**** LTA (lipoteichoic acid) is synthesized extracellularly by the LtaS enzyme (it is not a secreted product). MeeF/Y are homologous to ion transporters, are often regulated by Mn-inducible riboswitches, and have been shown to function in Mn export from cells both in *B. subtilis* (PMID: 31685536) and more recently in *E. coli* (PMID: 37214827). LTA polymers are not secreted, but are assembled on the cell surface, where they indeed function in metal buffering (lines 326-329).**

Could the inhibitory phenotypes related to zinc be a consequence of known inhibitory effects of zinc on the mechanisms of manganese import as documented by Chris McDevitt and co-workers?

**** We believe the referee is referring to the combined effect of Zn and the LtaS inhibitor (1771) as reported in Figure S9. These results demonstrate that Zn intoxication is specific to cells that are deficient in LTA synthesis (through mutation of one or more LtaS isozymes) and chemical inhibition of LtaS by 1771. This is unrelated to mechanism of inhibition cited in Chris McDevitt and coworkers, which refers to the ability of Zn to competitively**

inhibit Mn uptake through the MntABCD ABC transporter (PMID 22072971). This mechanism is not operative in *B. subtilis* since the major Mn importer is MntH, which is not inhibited by Zn. Instead, high Zn interferes with the electron transport chain as shown in *B. subtilis* (PMID: 27935957) and in *E. coli* (PMID: 7557331).

line 251. Metalation of CucA was subsequently shown to require two copper transporting ATPases and a cytosolic metallochaperone, with CucA secretion being absent in mutants lacking these proteins suggesting that metalation is assisted and co-coincident with secretion (J Biol Chem 2010 285: 32504–32511).

**** Thank you for this clarification. We have limited our discussion of the Mn-dependent cupin (MnCA).**

The use of "suggest" and "possibly" in the closing sentence: 'These results "suggest" that TerC proteins are important in both bacteria and eukaryotes for the proper functioning of exported proteins, "possibly" by mediating the co-translocational insertion of metals into nascent proteins during transit across membranes' is more cautious than elsewhere and perhaps some statements need to be toned-down and/or caveats added.

**** The Introduction and Discussion sections have been re-organized and expanded to clarify our conclusions (see referee 2, first comment).**

Reviewer #2 (Remarks to the Author):

The manuscript by He and colleagues describes a novel role for TerC proteins in the delivery of manganese to proteins exported via the SEC translocase. The findings built on their own analyses which defined TerC as manganese export systems, and the authors introduce these proteins as possibly functionally conserved across multiple kingdoms of life. The work performed here is of a high standard and the manuscript is very clearly written. The experiments performed here are logical and provide the necessary information required to draw conclusions regarding the proposed novel role for TerC in *B. subtilis*. However, the brevity of the manuscript presents a limitation in terms of drawing broader conclusions across bacteria and definitely when considering other kingdoms of life.

**** We appreciate the referee's supportive assessment. We have amended the text to expand and clarify our Discussion and further develop those ideas that may not have been clear due to the brevity of our presentation.**

Main comments:

This work may benefit from some *in silico* work, including the genetic organisation and conservation across the species, genus, between other bacteria and across kingdoms. Structural models are likely to assist too. These don't need to be described in detail in the main body of the text, but may appeal to those that are not in the field of *Bacillus* research.

**** We have added a new panel (Fig. 4a) to highlight the phylogenetic relatedness of the proteins tested in Fig. 4. We agree that a much broader *in silico* analysis will be valuable, although considerable efforts of this type are already published. Protein sequence similarity has been used to argue that the bacterial TerC proteins are part of a larger family of ion transporters as discussed in prior *in silico* work and now cited (lines 283, 340): "TerC proteins (Pfam03741) are a subgroup of the lysine exporter (LysE) superfamily of transporters and have seven TM segments and a conserved metal-binding site^{14,70}" and "Functional studies have linked diverse TerC and related UPF0016 proteins to the transport of Mn and Ca^{14,19,87,88}." Readers interested in the conservation across species and kingdoms can refer to these detailed analyses and the public resources that describe PFAM/UPF protein alignments and families. Further insights into functional conservation requires further study in diverse systems, and experimental determination of structures and mapping of the ion channels in model proteins.**

Expanding upon this, I would like to learn more about why MeeF appears to have a slightly stronger functional role than MeeY, and why more interacting proteins were identified. Other than the 40% similarity, what's different: expression, structure, distribution across the cell surface?

**** Our current model for MeeF/Y function is that these proteins allow the transit of Mn from the cytosol to the cell surface, where the ion is held by predicted metal-binding sites on extracytoplasmic loops. These loops may allow for specific delivery of Mn to client proteins as they begin to fold upon exit from the secretory channel. While MeeF/Y are redundant in function for many of the phenotypes noted, they are not identical in function (as**

the referee notes). Indeed, for some phenotypes we believe that there are dramatic differences in role, which provide a tool for better understanding the differentiation of roles for these two proteins in future studies. Our current thinking regarding the biological significance of having two paralogs is now further explained in the Discussion section (line 288-300).

Why is YjbE not contributing to the processes described here, and how may it fulfill its role during sporulation? Please discuss.

**** The third paralog (YjbE) is selectively expressed during sporulation according to Nicolas et al. as cited. While this protein plays a minor role under the conditions studied here, we do believe that it is likely more important during sporulation. We have added new data in Fig. 1a that demonstrate that YjbE is not contributing to fitness under the conditions studied here. YjbE is now discussed explicitly in the discussion section (see comment above).**

How are TerC proteins and their role in metalation of secreted proteins impacted by Mn starvation, something that bacterial pathogens readily encounter during infection?

**** The phenotypes described here are observed under conditions of relatively low Mn availability. LB medium has just enough Mn to support growth, and this is probably one reason why we observed the phenotypes reported. As host cells limit Mn during infection (e.g. through the action of calprotectin) the role of TerC family proteins is selectively directing Mn to surface enzymes (e.g LtaS) will become even more critical than for cells growing in the presence of excess Mn. This is now discussed (l. 300-303).**

Minor comments:

Line 64: Please don't specify the name of the mutant in the introduction, leave that until the results, where it has already been clearly stated too.

**** The mutant name was deleted from the line as requested. The mutant strain is described at the beginning of results.**

Line 75: This statement is superfluous, as this can be derived from the intro.

**** The statement was deleted.**

Line 643: Is it clearer bands, or simply more bands?

**** The statement in question, "Higher protease activities correspond to clearer bands on the gel matrix." was deleted since it is not needed. Samples with greater protease activity may have more bands, greater clearing (activity) in each band, or some combination.**

Line 101: Listing the molecular weights of the proteins seems confusing and Fig 1d doesn't actually show the specifics of those proteases listed, only that of the delta7 strain. Hence, denoting specific candidates seems appropriate for Fig S2 alone.

**** We agree and have removed the molecular weights as suggested for Fig. 1d, referring instead to Fig. S2 as suggested.**

Line 654: Please indicate total samples (were multiple biological replicates included on one day, as I assume that technical replicates are not displayed)

**** Figures have been revised so that each point represents a separate biological replicate, with the values shown being the average of technical replicates. This is now explained in the figure legends and the original data are in the provided Source Data file.**

Line 212: Please use these references for the strains with FLAG tag, as it may cause confusion.

**** We do not see strains mentioned at this point in the text. However, we have reviewed the text carefully to make sure that all references to strains are unambiguous.**

Line 138-139: This should be the other way around.

**** We have amended the sentence for clarity.**

Line 803: This sentence seems incomplete.

**** We have re-written the figure legend for clarity.**

Line 145: Please define that the MeeF interactome is broader than that of the MeeY.

**** We have added a sentence to address this point (l. 157).**

Line 811-812: I would say that the pattern is similar, it's the difference in intensity that is more striking.

**** We agree and have amended the text to clarify this point.**

Lines 183-185: Bit odd to explain what they do after already defining the result, restructure this order.

**** We have rewritten for clarity.**

Fig S8: Relative transcript to what? Scale is confusing. Why only two reps?

**** This experiment now has 4 biological replicates shown and we have clarified that all values are normalized to the *gyrA* transcript.**

Line 209: Please ascertain this transcriptionally.

**** We did not mean to imply transcriptional activation. We have changed the text (l. 229) to say "due to increased activity of the YqgS synthase."**

Lines 229-236: This really needs bioinformatics to support the choice of those two candidates and the claim that it's conserved.

**** A phylogenetic tree is added as panel 4a to provide context for the selection of these orthologs.**

Lines 239-246: This can be condensed.

**** We have edited this paragraph for brevity.**

Line 259: delete "its".

**** Corrected**

Line 266: Is Mn structural or any role in folding, as lack of metalation seems like an insignificant reason to not release the protein by the translocase? Or perhaps degradation is favourable as the protein would be non-functional, please discuss.

**** The discussion section has been revised to more fully describe the proposed role of TerC proteins in exoenzyme metalation.**

The legends of Fig 2D and S2 appear contradictory regarding the role of AprE.

**** The data in S2 suggest that AprE is required for the 17 kD band, assigned as a degradation product of Bpr. This band is still present in the FY mutant in Fig 1d, suggesting that there is at least some AprE secretion, as seen also in the immunoblot analysis of Fig. 2d. Thus, there is no contradiction.**

Reviewer #3 (Remarks to the Author):

This MS addresses the important issue of the metalation of extracytoplasmic proteins. This is important because many secretory Gram-positive proteins are metalloproteins or require metal ions for folding following translocation across the membrane (via the Sec translocase) in an essential unfolded/unstructured form. Slowly folding proteins are subject to proteolysis by quality control proteases and metal ions are known to act as folding factors, speeding up the folding at least for some proteins. This MS provides a mechanism for the metalation by the lower affinity metal ion Mn, so as to potentially optimise folding and avoid mis-metalation by higher affinity metal ions. There is a suggestion that folding provides an energy force that helps "pull" for the protein from the translocase and so metalation may help in this process. The current view is that metalation occurs in most cases via the concentration

(cf the medium) of metal ions in the negatively charged cell wall. However, this does not allow for mis-metalation for lower affinity metal ions. Therefore, the hypothesis proposed in this manuscript is attractive. I do, however, have some comments on what I see as potential experimental limitations that the authors should address. These are indicated below.

**** We thank referee for their comments. We agree with the referee that metal ions buffered by the anionic cell wall provide one source for metalation of nascent proteins. Further, metal ions (and often Ca, specifically) are often invoked as folding factors for extracellular enzymes. The role of LTA in metal buffering is now explicitly discussed in the text (lines 326-329).**

L104/Fig 1c/d

The legend states that "Supernatants were collected from ONCs with the same cell number(s)". What medium was used and did all the cultures reach stationary phase with the same numbers of cells and at the same time? I might have expected the FY mutant to have lower numbers or to have a slower growth in some media. I am also cautious about simply using data derived from overnight cultures (ONCs) without considering growth kinetics. Bacillus continues to produce many secretory enzymes in stationary phase and therefore cells that enter stationary phase earlier than others could generate more enzyme during their longer stationary phase. Therefore, the lower protease activity from the mutant could be due to a slower growth rate (ie reaching stationary phase later) and a resulting shorter time in stationary phase. Has this been taken into account?

**** Yes, this has been taken into account. As noted, protein secretion is mostly a stationary phase phenomenon with proteins accumulating for many hours after cells enter stationary phase. Under our conditions (LB) all strains used grow at very similar rates in liquid cultures as seen in Fig S1B. (This is distinct from the slow growth phenotype reported on solid medium, where the diffusion of nutrients is limiting and optimal growth requires the action of secreted proteases digest the peptides present in the medium). This is clarified where relevant in the text.**

L 107

The same point as above, although I suspect the interpretation is correct. The length of time in stationary phase could significantly affect secretory protein production and a better experiment would have been to take samples at a fixed time following the transition from exponential to stationary phase. The >5-fold reduction in extracellular Mn is to be expected from previous work on these genes as Mn exporters.

**** As noted above, Fig. S1B demonstrates identical growth kinetics so the time in stationary phase is the same. We agree that the reduction in Mn in the mutants is consistent with the role of these proteins as Mn exporters.**

L117

As above, and AmyQ (which requires Ca ions for folding) certainly accumulates in stationary phase.

**** As above, samples were taken at a fixed time following stationary phase since growth rates are so similar.**

L127

It is well known that AmyQ induces secretion stress as compared with native enzymes. As a Ca requiring enzyme it is not clear what the significance of these findings are. I think the most significant finding here is that secretion stress is not induced in the FY mutant, which I find surprising if Mn requiring enzymes are unable to fold/fold rapidly.

**** As the referee correctly notes, high levels of AmyQ induces the secretion stress response. However, in the FY mutant strain the secretion of AmyQ is compromised (Fig. 2e), an effect we attribute to translocon jamming. Since the level of secreted protein is decreased, one would not expect induction of the secretion stress response.**

L142

It could be argued that the results in Table 1 indicate that the MeeF and MeeY (membrane proteins themselves) simply interact with other membrane proteins. I don't see a clear justification, based on these data, for claiming MeeF and MeeY appear to function as part of the secretosome.

**** We understand the point raised but we respectfully disagree in light of the data that support a colocalization with components of the secretosome. The exact composition of the secretion complex is still under investigation,**

but most papers agree that there are core components of the secretion apparatus that interact reversibly with accessory factors such as YidC (for membrane proteins), foldases, and even the F1Fo ATPase. These are precisely the proteins we find in the Co-IP studies, whereas the many other membrane complexes and proteins are not detected. We have added a footnote to the relevant data table (Table 1) to point out that the membrane proteins we detect are a selected subset (12/40) of the most abundant membrane proteins.

We have also added citation to a prior study (Turkovicova et al.; PMID 26778143) that looked at the interactome of *E. coli* TerC using both Co-IP and blue native PAGE. Inspection of their results (in SI datasets) reveals that the complexes they see in blue native PAGE include TerC(A1x) together with many of the same secretosome components we identified.

L156

Same point as above about ONCs and secretory proteins.

**** The referee raises the point that perhaps the decrease in secreted proteases in the *ftsH* mutant strain results from slower growth and therefore a reduced amount of time in stationary phase. Under our conditions, the WT and the *ftsH* mutant strain grow similarly and reach stationary phase at about the same time, as now shown in Fig. S1b**

L168

Despite my comments about the way protease levels were estimated, the potential role of MeeF and MeeY in providing Mn as a folding factor helping to “pull” proteins from the translocase would be consistent with the role of FtsH clearing the translocase in the FY mutants.

**** We agree that the strong epistasis observed with the *ftsH* mutation is supportive of translocon jamming when metalation of nascent proteins is impaired.**

L204

The data on Mn addition to the medium in relation to LtaS seem clear. However, the medium (as indicated above) already contains some Mn that, in the wild type, would be at a higher concentration at the membrane/wall interface due to its mobile interaction with the phosphate in LTA. Could the influence of Mn addition on LtaS activity in the FY mutant be simply the result of controlling the level of Mn added to the medium to “just enough” (as indicated above and the influence on the WT as indicated below) for the wild type but not enough for the mutant. If LtaS folding was facilitated simply by the level of Mn in the medium, and this was lower in the FY mutant due to Mn retention by the cell, then the rate of folding of LtaS at lower Mn concentrations could slower, leading to its removal by quality control proteases. Additional Mn allow this to recover LTA synthesis.

**** We think this situation is complex, and that some proteins (e.g. LtaS) may be folded and released into the membrane even in the absence of metalation. The inactive LtaS can then be metalated by simply increasing Mn availability in the medium. For some other proteins, including perhaps metalloproteases, an inability to metalate the nascent protein may lead to translocon jamming. Unfortunately, there are no clearly defined assays for the phenomenon of translocon jamming, nor robust mechanisms for identification of “jammed” substrates. Addressing these challenges will require further study. However, we have more fully described the model for the role of TerC proteins in protein metalation in the Discussion.**

L264

From the data presented I think it is too early to argue that MeeF and MeeY are accessory subunits of the holotranslocon.

**** We understand the concern. However, the data we present based on Co-IP, defects in secretion, and epistasis with *ftsH* are consistent with the types of data used previously to assign other proteins as accessory subunits (part of a larger secretosome complex) that forms around the holotranslocon. This conclusion is also supported by prior work on *E. coli* TerC (Turkovicova et al.). Further, the *Arabidopsis* homolog (AtTerC) also interacts with a YidC homolog (ALB3) (Schneider, A. et al. 2014), consistent with our model.**

REVIEWERS' COMMENTS

Reviewer #1 (Remarks to the Author):

To address the first question raised in the initial review, the authors should add a sentence at a prominent location in the manuscript to make readers aware that:

"Future studies will be needed to establish how a weak binding metal such as Mn(II) is retained by the active sites of enzymes located outside the plasma-membrane post metalation by MeeF/Y: Possibilities include oxidation to less exchangeable Mn(III), kinetic trapping post folding or repeated insertion by MeeF/Y."

Inclusion of such a statement is important if the paper is published in Nature Communications with its broad readership, many of whom may have limited knowledge of bioinorganic chemistry and may be left with the impression that the delivery of Mn(II) once to an active site located outside the plasma-membrane (such as in LtaS) fully resolves the challenge (set out in the introduction and opening paragraphs of the discussion) to metalate exoenzymes with lower affinity metals in such a location where buffered metal concentrations (activities) are not controlled.

Reviewer #2 (Remarks to the Author):

Excellent work team. The revised version of the manuscript adequately addresses my comments and requests from the previous round of review.

Reviewer #3 (Remarks to the Author):

This revised manuscript improves and clarifies aspects of the original. The work is novel and the experimental work is appropriate and justifies the conclusions. It represents significant contribution to an important component of the protein secretion pathway, by providing a mechanism for the metalation of proteins on the trans side of the membrane with Mn. This manuscript is of broad significance, and particularly important in relation to the essential process of secretion and the exploitation of this process for the production of industrial enzymes.

REVIEWERS' COMMENTS

Reviewer #1 (Remarks to the Author):

To address the first question raised in the initial review, the authors should add a sentence at a prominent location in the manuscript to make readers aware that:

"Future studies will be needed to establish how a weak binding metal such as Mn(II) is retained by the active sites of enzymes located outside the plasma-membrane post metalation by MeeF/Y: Possibilities include oxidation to less exchangeable Mn(III), kinetic trapping post folding or repeated insertion by MeeF/Y."

Inclusion of such a statement is important if the paper is published in Nature Communications with its broad readership, many of whom may have limited knowledge of bioinorganic chemistry and may be left with the impression that the delivery of Mn(II) once to an active site located outside the plasma-membrane (such as in LtaS) fully resolves the challenge (set out in the introduction and opening paragraphs of the discussion) to metalate exoenzymes with lower affinity metals in such a location where buffered metal concentrations (activities) are not controlled.

We added the relative sentences in the paper (line 335):

"Future studies will be needed to establish how weak binding metals such as Mn are retained by exoenzymes. For some proteins, Mn may be oxidized to less exchangeable Mn(III), or the bound Mn may be kinetically trapped after protein folding. Alternatively, MeeF and MeeY may be able to repeatedly load metal into those proteins that are retained in the membrane (LtaS) or in the vicinity."

Reviewer #2 (Remarks to the Author):

Excellent work team. The revised version of the manuscript adequately addresses my comments and requests from the previous round of review.

We really appreciate former comments and suggestions.

Reviewer #3 (Remarks to the Author):

This revised manuscript improves and clarifies aspects of the original. The work is novel and the experimental work is appropriate and justifies the conclusions. It represents significant contribution to an important component of the protein secretion pathway, by providing a mechanism for the metalation of proteins on the trans side of the membrane with Mn. This manuscript is of broad significance, and particularly important in relation to the essential process of secretion and the exploitation of this process for the production of industrial enzymes.

Thank you for these comments!